# Association of Knowledge and Health Habits with Physiological Hydration Status

**DOI:** 10.3390/nu16111541

**Published:** 2024-05-21

**Authors:** Brendon P. McDermott, Xiujing Zhao, Jennifer C. Veilleux

**Affiliations:** 1Department of Health, Human Performance, and Recreation, University of Arkansas, Fayetteville, AR 72701, USA; xiujingz@uark.edu; 2Department of Psychological Science, University of Arkansas, Fayetteville, AR 72701, USA

**Keywords:** health index, hydration status, knowledge, fluid intake, urine

## Abstract

The association of hydration knowledge and health habits with hydration status and fluid intake is rarely examined. We sought to determine whether knowledge or physical health behaviors predict physiological hydration status and fluid intake. Ninety-six participants (59 female; 27 ± 10 year) completed the previously validated hydration survey. Participants then recorded total fluids consumed (TFC), collected urine, and tracked void frequency for 24 h. Hydration status was assessed via 24 h urine specific gravity (USG) and osmolality (U_osm_). Health behaviors included self-reported physical activity, BMI, smoking, alcoholic drinking, and sleep status. TFC was significantly correlated with 24 h USG (r = −0.390; *p* < 0.001), U_osm_ (r = −0.486; *p* < 0.001), total urine volume (r = 0.675; *p* < 0.001), and void frequency (r = 0.518; *p* < 0.001). Hydration knowledge was not correlated with 24 h USG (r = 0.085; *p* = 0.420), U_osm_ (r = 0.087; *p* = 0.419), urine total volume (r = 0.019; *p* = 0.857), void frequency (r = 0.030; *p* = 0.771), or TFC (r = 0.027; *p* = 0.813). Hydration knowledge did not predict 24 h USG (LR^+^ = 1.10; LR^−^ = 0.90), U_osm_ (LR^+^ = 0.81; LR^−^ = 1.35), or TFC (LR^+^ = 1.00; LR^−^ = 1.00). Health habits did not predict 24 h USG, U_osm_, or TFC. In conclusion, self-reported 24 h diet and fluid log recording is comparable to hydration status verification via 24 h urine collection. Hydration knowledge and health habits are not related to, or predictive of, hydration status.

## 1. Introduction

Water is an essential nutrient for optimal physiological function [1,2]. Euhydration fosters acute and chronic health [3] and plays a crucial role in metabolism, homeostasis, thermoregulation, and circulatory function [2]. Hypohydration is associated with myriad physiological and psychological consequences, such as impaired health [4], a deficit in cognition and mood [5], and decreased exercise performance [6,7]. The US National Academy of Medicine (NAM) recommends adequate total daily fluid intake (sum of drinking water, all beverages and food moisture) for adult men and women as 3.7 L/day and 2.7 L/day, respectively [8]. However, data from the 2005–2009 National Health and Nutrient Examination Survey [9] shows that less than 10% of adults meet these guidelines, regardless of sex. A survey study in China found that approximately 32% of 1483 adults from four cities drank less fluid than the amount recommended by the Chinese Nutrition Society in 2007 (1200 mL/day) [10]. A cross-sectional survey among adults in 13 countries reported that less than 50% of the women and 60% of the men do not comply with European Food Safety Agency (EFSA) guidelines (1.6 L/day for women; 2.0 L/day for men + 20% from food) for adequate fluid intake [11].

Daily fluid intake is an effective, direct strategy to pursue an optimal hydration status (i.e., euhydration) [12]. Previous studies [13,14,15] reported strong associations between self-reported daily fluid intake and 24 h urinary biomarkers (i.e., urine color, urine volume, urine specific gravity, and urine osmolality) for hydration status verification. Fluid intake is related to exercise intensity [16], environmental conditions (temperature, humidity, and wind speed) [17], and fluid flavor [18]. However, the association of fluid intake with other determinants, such as hydration knowledge and healthy behaviors, is not clear. It would be helpful to understand and identify factors influencing fluid intake so that appropriate interventions could improve individual hydration status and fluid balance.

Hydration knowledge is a potential factor impacting fluid intake. Previous studies investigating the association of hydration knowledge with fluid intake in athletes report mixed findings [19,20,21,22,23]. There was no significant relationship between hydration knowledge and fluid intake in football, weightlifting, or endurance athletes [19,20,23]. In contrast, other studies reported hydration knowledge was associated with fluid intake in collegiate track and field athletes and soccer athletes [21,22]. Compared to research on athletes, few studies [23,24] examine hydration knowledge among the general population. Poor hydration knowledge on daily intake of water was observed among adults in the UK, France, Spain, and Saudi Arabia [23]. Veilleux et al. [24] elucidated no association between hydration knowledge and fluid intake in the US.

Health-related behaviors are recordable variables that may impact daily fluid intake. Physical activity should result in increased fluid intake based on replacement of sweat losses from evaporation [25]. The American College of Sports Medicine [26] recommends 150 min of moderate-intensity physical activity every week. Previous studies [27,28] suggest that meeting ACSM guidelines is associated with significantly more fluid intake. Beyond physical activity, the association of other health-related behaviors such as body mass index (BMI), smoking, sleeping, and alcohol drinking with fluid intake is unclear. To our knowledge, only one study [27] evaluated whether smoking, sleeping, or BMI are associated with fluid intake. They found that only smoking status is related to fluid intake; former smokers drink more water than those who never smoked. A comprehensive understanding and quantitative measurement of how hydration knowledge and health-related behaviors are related to fluid intake among the general population could be used to improve overall health and hydration status. Previous studies have suggested continual investigation in this area to improve our knowledge of health behavior integration.

Our purposes were threefold; first, we examined the relationship between self-reported fluid intake and a diet log to physiological hydration status. Since fluid intake is directly related to hydration status, we investigated the relation between self-reported fluid intake and hydration biomarkers. Second, we aimed to evaluate the association of hydration knowledge with hydration status. Lastly, we sought to determine whether physical health-related behaviors (physical activity, smoking, sleeping, alcohol drinking, BMI) predict physiological hydration status or fluid intake. Our hypotheses were that hydration status would be directly linked to self-reported fluid intake, but not with hydration knowledge. Also, we hypothesized that health habits would predict hydration behaviors; healthier participants would have significantly improved hydration stats and total fluid intake.

## 2. Materials and Methods

### 2.1. Description of Participants

The present mixed-methods study utilized cross-sectional data on hydration-related knowledge, health-related behavior, and self-reported total fluid intake among adults. Ninety-six participants (female: 59, male: 37; age range 19–54 year) were involved in our study. Our convenience sample of participant demographics is included in Table 1. Before recruiting participants, this study was approved by the University of Arkansas Institutional Review Board. Participants were recruited via campus e-mail, verbal announcements at group meetings, and class advertisements. Inclusion criteria for this study required participants to be >18 year, free from chronic disease, not taking prescription medication that alters kidney function or fluid balance, and self-identified as otherwise healthy. All participants reported to our lab to complete our survey and initially provided informed consent electronically. Prior to signing the informed consent form, participants were instructed to read it thoroughly and encouraged to ask clarification questions about the purpose, risks, benefits, time commitment, and procedures of participation. Participants were free to withdraw from the study at any time. Data were collected between October 2020 and April 2022.

### 2.2. Assessment of Daily Fluid Intake, Hydration Knowledge, and Status

Diet and Fluid Consumption Logs. Diet and fluid consumption logs have been previously used to collect information on food and fluid consumption [11,14,15,29,30]. Participants were provided diet and fluid consumption logs to record, in detail, food and fluid consumption over the same 24 h period as the urine collection. Researchers educated participants on data recording responsibilities and answered questions prior to participants leaving the lab. Total fluid intake included total fluid consumed and dry food water content and was quantified via nutrition software (Nutritionist Pro version 7.8.0, Axxya Systems, Redmond, WA, USA).

Hydration knowledge was assessed via a previously content validated 16-item hydration knowledge scale [24]. This survey was administered online via desktop computer when participants were in our lab. The hydration knowledge survey assessed the accuracy of participants’ hydration knowledge.

Twenty-four h urine collection. Upon completion of our online survey, participants were provided a urine jug to collect total urine output over a 24 h period. Researchers assessed and recorded 24 h urine specific gravity (USG), urine osmolality (U_osm_), urine volume, and void frequency [31] immediately after the urine jug was returned. Hydration status was assessed via 24 h USG (Master-SUR, Atago Co., Ltd., Tokyo, Japan) and freezing point depression, U_osm_ (3250, Advanced Instruments Inc., Norwood, MA, USA) [32,33]. Void frequency was assessed as described in previous investigation [31].

### 2.3. Health-Related Behavior Recoding and Assessment

Health-related behaviors included self-reported physical activity, weight (BMI), smoking, alcoholic drinking, and sleep status. Physical activity was assessed using the International Physical Activity Questionnaire (IPAQ) [34]. IPAQ data were recorded as a binary variable indicating whether or not the participant was meeting the ACSM physical activity guidelines. BMI was calculated from height and body mass recorded in our lab, then recorded as normal (BMI < 25.0 kg/M^2^) or overweight and obese (BMI > 25.0 kg/M^2^). Smoking status was recorded as smoking or nonsmoking. Alcoholic drinking status was recorded as light drinking (<2 times/week) or heavy drinking (>2 times/week). Sleeping status was recorded as the number of hours of sleep on an average night <6 h or >6 h. Health behaviors were then recoded as healthy or unhealthy prior to analysis.

The five health-related behaviors were recoded as healthy (labeled as 1) and unhealthy (labeled as 0) prior to analysis. Our cut-points and classification were similar to a previous study [28]. The healthy or unhealthy index scores were the sum of the five health-related behaviors after recoding. The health index scores ranged from 0 to 5. The health index scores of 3, 4, and 5, or 4 and 5, were defined as healthy habits. In contrast, the health index scores of 0–2 were defined as unhealthy. We chose to analyze healthy and unhealthy habits based on our small sample size (41 of our participants met 3 healthy habit criteria) and to identify the strength of association between our variables. Therefore, we included two recoding strategies for healthy habits.

### 2.4. Statistical Analysis

Data were analyzed using SPSS (v. 28) software for Windows (SPSS, Chicago, IL, USA) and Excel (2016) for Windows (Microsoft, Redmond, WA, USA). Descriptive characteristics were calculated for all study variables and reported as mean ± standard deviation (SD). Differences between the female and male groups were assessed with independent t-tests. Spearman rho correlation was used to report the association of hydration knowledge score with hydration status and fluid intake [35]. Standard crosstab analysis with likelihood ratio calculations was conducted to assess whether hydration knowledge and health habits predict hydration status and fluid intake. Prior to conducting predictive analyses, the cut-points were as follows: hydration knowledge scores (16), USG (1.020), U_osm_ (500 m_osm_/L), total fluid intake (women: 2.7 L/day, men: 3.7 L/day), and health habits total health index score ≥3 or ≥4, respectively. Statistical significance was set a priori at *p* ≤ 0.05.

## 3. Results

### 3.1. Characteristics of Participants and Sex Differences in Hydration Status Variables and Fluid Intake, and Hydration Knowledge Scores

The final sample comprised 96 participants involved in the current study. Table 1 provides the participant demographic characteristics. Table 2 presents sample health-related behavior characteristics. Table 3 presents hydration variable differences between females and males including 24 h USG, U_osm_, total urine volume, void frequency, total fluid consumed, and hydration knowledge scores.

**Table 1 nutrients-16-01541-t001:** Sample characteristics in data collection.

Characteristics	
Age (y)	27 ± 10
Women	59 (61.5)
White	75 (78.1)
Employment	
Unemployed or student	60 (63.2)
Employed	35 (36.8)
Height (m)	1.68 ± 0.09
Mass (kg)	71.36 ± 15.84
Weight status	
Normal weight (BMI < 25)	52 (54.2)
Overweight (BMI 25–30)	30 (31.2)
Obese (BMI > 30)	14 (14.6)

Data are presented with mean ± SD or n (%).

### 3.2. Total Fluid Consumed and Hydration Status

Table 4 presents 24 h total fluid consumption related to 24 h USG, U_osm_, urine volume, and void frequency. According to the scale of the Spearman correlation coefficient (*ρ*) [33], 24 h total fluid consumed had a low correlation with 24 h USG, moderate correlation with 24 h U_osm_ and void frequency, and strong correlation with 24 h urine volume.

### 3.3. Hydration Knowledge, and Status, and Total Fluid Intake

Twenty-four h total fluid consumed for our sample was 3288 ± 1515 mL. The mean hydration knowledge score was 16 ± 6 out of 32. For hydration status, 24 h urine volume was 1933 ± 990 mL; 24 h U_osm_ was 436 ± 210 m_osm_/L; 24 h USG was 1.012 ± 0.006, and 24 h void frequency was 7 ± 3 times.

Table 5 presents the hydration knowledge correlation with hydration status variables. Table 6 shows that the hydration knowledge score did not predict hydration status variables (USG and U_osm_) and total fluid intake.

### 3.4. Health Habits Predicting Hydration Status and Total Fluid Intake

Table 7 presents the prediction of hydration status and total fluid intake by health habits. It also suggests that health habits (healthy index scores of 3 and 4) did not predict hydration status variables (USG and U_osm_) or total fluid consumed.

## 4. Discussion

The main finding of our study was that self-reported fluid intake is comparable to 24 h urine collection for hydration status verification. Confirming previous findings relating knowledge and behavior, we identified that hydration specific knowledge was not associated with physiological hydration status or fluid intake. We did identify preliminary evidence that a health behavior index was mildly predictive of hydration status and fluid intake.

### 4.1. Total Fluid Consumed Was Related to Physiological Hydration Status

Self-reported total fluid consumed was a determining factor in physiological hydration status [24]. Assessing the link between fluid intake and hydration status may provide crucial information on daily fluid intake to help individuals maintain euhydration and avoid health consequences related to hypohydration [13]. Previous studies [36,37] suggest a linear correlation between fluid intake and biomarkers related to hydration status, which was consistent with the current study. We found that self-reported 24 h total fluid consumed was significantly associated with 24 h USG, U_osm_, urine volume, and void frequency. Perrier et al. [13] illustrated strong associations (*r* = 0.6; *p* < 0.001) between total fluid intake and 24 h U_osm_, urine color, USG, urine volume, and solute concentrations in healthy adults. Zhang et al. [14] and Zhang et al. [15] investigated the association of total fluid intake with urine biomarkers that related to physiological hydration status among young adults, with a sample similar to ours, and found strong relationships between daily total fluid intake and 24 h urine biomarkers among young adults, especially for 24 h urine volume and osmolality. In contrast, the current study found only a strong correlation between 24 h urine volume and total fluid intake, not in urine osmolality. One reason might be that the cut-point for urine osmolality indicating euhydration was not identical. The two previous studies [14,15] used U_osm_ 800 m_osm_/L, while the current study used 500 m_osm_/L. We did this to reflect a more aggressive classification of hydration status [33]. Despite these differences, our data are similar to the previous literature.

### 4.2. Hydration Knowledge Was Not an Adequate Predictor for Physiological Hydration Status or Fluid Intake

Our study attempted to assess the criterion validity for a previously content validated hydration survey. This would be valuable as an indicator for large populations when 24 h urine collection and analysis are not feasible. The average knowledge score in the present study was 16 ± 6, which was greater than previously reported [24]. Despite the average hydration knowledge scores being different, the results of the previous study indicated that hydration knowledge scores were not associated with fluid intake, consistent with our data. Our study suggests hydration knowledge scores are not associated with physiological hydration status and fluid intake. The current study also elucidates that the hydration knowledge score did not predict hydration status and fluid intake (Table 6). The findings can be explained by the low correlation between knowledge and behaviors: knowing the “right” thing to do may not always tranfer into “good” behaviors [30,38,39]. For example, people know smoking impairs health and induces disease, but this knowledge does not necessarily change behavior. Likewise, hydration knowledge in the present study did not guaratee that participants achieved adequate fluid intake. Overall, our sample demonstrated a well-hydrated cohort based on mean hydration data. This may hide the association between hydration knowledge and physiological hydration status and fluid intake.

### 4.3. Health Habits Did Not Predict Physiological Hydration Status and Fluid Intake

This is the first study to evaluate a healthy index using five health behaviors (smoking, sleeping, alcoholic drinking, weight, and physical activity status) in predicting hydration status or total fluid intake. The present study suggests that these health habits are not predictive of hydration status or fluid intake. Although no previous studies have assessed whether combined health habits, using multiple health behaviors, predict hydration status and fluid intake, previous research [27] has investigated the association of smoking, sleeping, BMI, and physical activity with fluid intake. They found greater fluid intake was associated with over 150 min moderate-intensity physical activity per week and former smoking status. It is reasonable that those who complete more physical activity drink more. The interesting finding in the previous study was that former smokers get more water than non-smokers, which might be explained by tobacco cessation program encouraging participants to drink more water. Former smokers may build water intake habits during tobacco cessation programs and carry them further following the program. In the present study, healthy habits did not predict hydration status or total fluid intake. As mentioned, our sample was well-hydrated, which might impair the relationship between health habits and hydration status.

### 4.4. Limitations and Future Study

The first limitation of this study was the small sample size, which may prevent the findings from being extrapolated. At the same time, our sample was well-hydrated (exceeding recommended guidelines on average), which does not match epidemiological data. A larger sample size with physiological data would allow for more extensive predictive regression analysis for more robust outcomes. Future study should seek more physical or psychological health variables related to hydration to predict hydration status and behaviors. It is plausible that our participants may have consumed alcoholic beverages only one or two days per week, yet they engaged in unhealthy behaviors (binge drinking). We chose to classify two or less days per week as ‘healthy’ based on assumptions that have limitations. Furthermore, investigating hydration status and behaviors of other populations such as children or elders is necessary. Our data were limited to young adults. Lastly, large representative samples are needed to assess the criterion validity for this hydration survey.

## 5. Conclusions

Our data show that self-reported 24 h diet and fluid log recording is comparable to hydration status verification via 24 h urine collection. We further validated that self-reported fluid logs track physiological hydration status. Our results are consistent with previous research in the limitation of knowledge related to healthy behaviors. Hydration knowledge is not related to hydration status or daily fluid intake. Changing hydration knowledge with educational interventions may not be an effective strategy to improve hydration behaviors. We also showed that a health index is not predictive of hydration status, meaning that individuals who exhibit improved healthy behaviors are not necessarily including adequate fluid intake in their daily habits. Future research should seek to evaluate potential links between survey data and hydration status in a more varied hydration population (varying ages, low and high drinkers).

## Figures and Tables

**Table 2 nutrients-16-01541-t002:** Sample health-related behavior characteristics.

	Health-Related Behavior Variables											
Sleeping Status: n (%)		Smoking Status: n (%)		Alcoholic Drinking Status: n (%)		BMI: n (%)		Physical Activity: n (%)		Health Index Score: n		
1	0	1	0	1	0	1	0	1	0	≥3	≥4	≤2
77 (80.2)	19 (19.8)	89 (92.7)	7 (7.3)	72 (75)	24 (25)	52 (54.2)	44 (45.8)	40 (41.7)	56 (58.3)	67	26	29

1 = healthy index, 0 = unhealthy index.

**Table 3 nutrients-16-01541-t003:** Sex comparison in hydration status variables, fluid intake, and hydration knowledge scores.

Variable	Men	Women	*p*-Value
24 h USG	1.013 ± 0.005	1.012 ± 0.006	0.142
24 h U_osm_	481 ± 189	407 ± 209	0.106
24 h total urine volume	1770 ± 792	2019 ± 1094	0.244
24 h void frequency	6 ± 3	7 ± 3	0.182
24 h total fluid consumed	3222 ± 1538	3303 ± 1520	0.822
Hydration knowledge scores	15 ± 6	16 ± 5	0.443

Data are presented as mean ± SD.

**Table 4 nutrients-16-01541-t004:** Correlation between self-reported fluid intake and physiological hydration status.

Variables	24 h Total Fluid Consumed	
	*ρ*	*p*-Value
24 h USG	−0.390 **	<0.001
24 h U_osm_	−0.486 **	<0.001
24 h total urine volume	0.675 **	<0.001
24 h void frequency	0.518 **	<0.001

** Correlation is significant at the 0.01 level (2-tailed).

**Table 5 nutrients-16-01541-t005:** Correlation between hydration knowledge and hydration status variables.

Variables	Hydration Knowledge	
	*ρ*	*p*-Value
24 h USG	0.085	0.420
24 h U_osm_	0.087	0.419
24 h total urine volume	0.019	0.857
24 h void frequency	0.030	0.771
24 h total fluid consumed	0.027	0.813

**Table 6 nutrients-16-01541-t006:** Predictive analysis for hydration status and fluid intake by hydration knowledge.

Variable	24 h USG (1.020)	24 h U_osm_ (500 m_osm_/L)	Total Fluid Consumed
Sensitivity	0.55	0.53	0.55
Specificity	0.50	0.34	0.45
LR+	1.10	0.81	1.00
LR−	0.90	1.35	1.00

Hydration knowledge was not predictive of USG, U_osm_, or total fluid consumed.

**Table 7 nutrients-16-01541-t007:** Predictive analysis for hydration status fluid intake by health habits (health index scores ≥3 or 4).

Variable	USG (1.020)		U_osm_ (500 m_osm_/L)		Total Fluid Consumed	
	HIS ≥ 3	HIS ≥ 4	HIS ≥ 3	HIS ≥ 4	HIS ≥ 3	HIS ≥ 4
Sensitivity	0.60	0.49	0.75	0.30	0.79	0.63
Specificity	0.40	0.80	0.43	0.79	0.37	0.61
LR+	1.01	2.45	1.32	1.45	1.25	1.63
LR−	0.99	0.64	0.58	0.88	2.19	0.60

HIS: health index score.

## Data Availability

The original contributions presented in the study are included in the article, further inquiries can be directed to the corresponding author.

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
