# Peer review of "Association of Knowledge and Health Habits with Physiological Hydration Status"

_nutrients, 2024, doi:10.3390/nu16111541_

Round 1

Reviewer 1 Report

Comments and Suggestions for Authors

Thank you for asking me to review this submission.

There is a lack of clarity in a number of areas, which I describe in the attached document.

Comments on the Quality of English Language

Good, minor revisions

Author Response

Thank you very much for taking the time to review this manuscript. Please find detailed responses below and the corresponding revisions/corrections highlighted in the re-submitted manuscript.

Reviewer 3

  • Inn title mention ages of the participants, and also study design, in-line with good reporting guidelines.

The authors have reviewed the author guidelines and previous tables of contents for the journal. It appears not customary, or required, to offer so much in the title of a published article. We have made efforts, based on reviewer comments, within the manuscript to clarify these points to be clear to the reader. We will defer to the section editor in determination of whether to include these within the title.

  • Mention mean ages of participants

Table 2 includes mean ages of participants and the remainder of subject demographics for our population. Our age range has also been added to line 88.

  • 1st paragraph: mention the EFSA fluid intakes (1.6L/day for women; 2.0L/day for men + 20% from food)

Per your suggestion, we have added this information in lines 40-41.

  • Throughout, could the authors make clear that they seem to be talking about adults <65 years, as the evidence is not quite the same for those >65years, or if talking about adults all ages –making that explicit too. Especially as this is such a young cohort in this study.

We have added our specific demographic age range and continually refer to our sample as ‘adults.’ We also added information within our discussion section. We hope that this clarifies a ‘younger’ population and data that would not necessarily reflect ‘older adults.’

  • Note that assessing hydration status and assessing fluid intake are not quite the same thing, so do make this clear. Initially (and especially first paragraph), the discussion is around fluid intake, and then in the last two paragraphs, the term ‘hydration status is used, and appears to be used interchangeably. I think you need a paragraph discussing the merits (or not) of measures of hydration status, as you are using them in this study as outcome measures. How valid and robust are they in this population? (note not at all in older adults, but this is not what this study is about!)

We have added information and altered sentences throughout to assure clarity between fluid intake and hydration variables. In hydration science, there is a plethora of data suggesting acceptability of any of our hydration variables (USG, osmolality, 24-hr urine output, void frequency). These are all standard measures of hydration within all published research in the field within the past 20 years for adults.

  • 2, line 79: ‘guessed’??!! perhaps use the term ‘hypothesised’ for a more international audience!

Per your suggestion, we have revised line 83.

  • Also, clarify what type odf dehydration is being discussed here. I am assuming ‘low-intake’?

We optioned to utilize hypohydration throughout the manuscript. Since dehydration, by definition, is a process of losing body water, it does not really apply. We are regarding hypohydration which, by definition, is a state of low body water.

  • I would advise differentiating between the two types of dehydration described by Thomas et al, 2008. He distinguishes between water-loss (more commonly called ‘low-intake’ now, also called intracellular) and hypovolaemia (extracellular). Also see Lacey et al, 2019 for more-up-to-date definitions.

In the current study, we used the term hypohydration that refers to a state condition of reduced total body water (Armstrong & Johnson, 2018). This terminology is consistent within hydration science. Specifically, there are differences, as the reviewer has pointed out. We were not regarding dehydration (water loss, or process), but are evaluating hypohydration (state). There are no acute changes involved within our study. We chose to refer to the citation below rather than citations offered as medical and differentiating acute changes in hydration based on water losses.

Armstrong, L. E., & Johnson, E. C. (2018). Water intake, water balance, and the elusive daily water requirement. Nutrients10(12), 1928.

  • Reporting guideline used? please mention.

The authors are unsure what is meant by this comment. Please clarify or provide a line for specific reference.

  • Move mention of study design to start of methods.

Per your suggestion, we have revised in lines 87-88 to include this information.

  • Describe sampling strategy: convenience?

Yes. Lines 90-95 should clarify.

  • Any power calculations for sample size?

We did not conduct a power analysis for our sample size. However, the current study was similar to or exceeded those of similar, published research (Decher et al.,2008; Perrier et al., 2013; Zhang et al., 2017).

Decher, N. R.; Casa, D. J.; Yeargin, S. W.; Ganio, M. S.; Levreault, M. L.; Dann, C. L.; James, C. T.; McCaffrey, M. A.; O’Connor, C. B.; Brown, S. W. Hydration Status, Knowledge, and Behavior in Youths at Summer Sports Camps. International Journal of Sports Physiology and Performance 2008, 3 (3), 262–278. https://doi.org/10.1123/ijspp.3.3.262.

Perrier, E.; Rondeau, P.; Poupin, M.; Le Bellego, L.; Armstrong, L. E.; Lang, F.; Stookey, J.; Tack, I.; Vergne, S.; Klein, A. Relation between Urinary Hydration Biomarkers and Total Fluid Intake in Healthy Adults. Eur J Clin Nutr 2013, 67 (9), 939–943. https://doi.org/10.1038/ejcn.2013.93.

Zhang, N.; Du, S.; Tang, Z.;  Zheng, M.; Yan, R.; Zhu, Y.; Ma, G. Hydration, Fluid Intake, and Related Urine Biomarkers among Male College Students in Cangzhou, China: A Cross-Sectional Study—Applications for Assessing Fluid Intake and Adequate Water Intake. International Journal of Environmental Research and Public Health 2017, 14 (5), 513.

  • Eligibility criteria were mentioned at the end paragraph, but Iwould advise restructuring of the paragraph so that the methods are reported in the order that they took place.

Per the suggestion, we have revised lines 90-95.

  • Dates of data collection?

This information has been added to lines 99-100.

  • How is ‘otherwise healthy’ defined?

Lines 93-95 have been edited to include specifics on this information.

  • How were participants taught/guided/educated in completing these logs, and who provided this information?

Lines 108-110 now include this information.

  • Were the food and fluid logs externally validated or created by the research team, and if so, how were they developed?

Food and fluid logs were previously used and validated. See citation below (Atkins et al., 2021). This has been added to the manuscript as well.

Atkins, W. C., McDermott, B. P., Kanemura, K., Adams, J. D., & Kavouras, S. A. (2021). Effects of hydration educational intervention in high school football players. The Journal of Strength & Conditioning Research, 35(2), 385-390. doi: 10.1519/JSC.0000000000003866.

  • Ditto re urine collection? How was voiding frequency recorded?

This urine collection procedure has been used for years without question. Void frequency collection was validated previously (Tucker et al., 2016). Participants label the urine jug when they urinate for 24 hours. This citation has been added.

Tucker, M. A., Gonzalez, M. A., Adams, J. D., Burchfield, J. M., Moyen, N. E., Robinson, F. B., ... & Ganio, M. S. (2016). Reliability of 24-h void frequency as an index of hydration status when euhydrated and hypohydrated. European journal of clinical nutrition, 70(8), 908-911.

  • How soon after the 24hr urine samples were returned were the other tests done, and what is the evidence that this timescale is OK, and no deterioration has occurred?

24-hr urine specific gravity (USG), urine osmolality (Uosm), urine volume, and void frequency were assessed and recorded immediately after the urine jug was returned.

This test procedure and timescale were similar to previous studies (Tucker et al., 2016; Zhang et al., 2021)

Tucker, M. A., Gonzalez, M. A., Adams, J. D., Burchfield, J. M., Moyen, N. E., Robinson, F. B., ... & Ganio, M. S. (2016). Reliability of 24-h void frequency as an index of hydration status when euhydrated and hypohydrated. European journal of clinical nutrition, 70(8), 908-911.

Zhang, J.; Ma, G.; Du, S.; Zhang, N. The Relationships between Water Intake and Hydration Biomarkers and the Applications for Assessing Adequate Total Water Intake among Young Adults in Hebei, China. Nutrients 2021, 13 (11), 3805. https://doi.org/10.3390/nu13113805.

Previous investigation suggests much longer stability of urine for assessment completion.

  • Who did these measurements?

Line 114 now included ‘researchers.’

  • Any validation tests for any of the participants to check accuracy of measurements?

Accuracy of measurements are reported in previous publications;

McDermott, B. P., Anderson, S. A., Armstrong, L. E., Casa, D. J., Cheuvront, S. N., Cooper, L., ... & Roberts, W. O. National athletic trainers' association position statement: fluid replacement for the physically active. Journal of athletic training 2017, 52(9), 877-895.

Armstrong, L. E. Hydration status: the elusive gold-standard. Journal of the American College of Nutrition 2007;26(5 Suppl):575S-584S.

  • How were the hydration knowledge surveys administered? Online of F2F, and if so, by whom? What is the max score, and what do lower scores mean?

Hydration knowledge surveys are administered online. A previous study (Veilleux et al., 2020) provided the measurement development and validation of the hydration knowledge scale. Furthermore, these procedures follow guidelines for scale development (Clark & Watson,1995; Hurley et al., 1997). The maximum score on the hydration knowledge scale is 32. The lower score refers to the individual who lacks knowledge-related hydration.

Veilleux, J. C., Caldwell, A. R., Johnson, E. C., Kavouras, S., McDermott, B. P., & Ganio, M. S. (2020). Examining the links between hydration knowledge, attitudes and behavior. European journal of nutrition, 59, 991-1000.

Clark LA, Watson D (1995) Constructing validity: basic issues in objective scale development. Psychol Assess 7:309–319. https:// doi.org/10.1037//1040-3590.7.3.309

Hurley AMYE, Scandura TA, Chester A et al (1997) Exploratory and confirmatory factor analysis: guidelines, issues, and alterna- tives. J Organ Behav 18:667–683

  • I think more details around classifications of light/heavy drinking and smoking/non-smoking needed, also sleep. Eg, if drinking is based on frequency – what about twice weekly binge drinkers, for example?

The classification of drinking, smoking, and sleep status in the current study is similar to that used previously (Goodman et al., 2007). In our data, like drinking status, we only collected how many times the participants drink per week. We don’t know whether they are binge drinkers or not. This is a great comment and we have added this to our limitations.

Goodman, A. B., Blanck, H. M., Sherry, B., Park, S., Nebeling, L., & Yaroch, A. L. (2013). Peer reviewed: behaviors and attitudes associated with low drinking water intake among US adults, food attitudes and behaviors survey, 2007. Preventing chronic disease, 10.

  • Some repetition around the description of the health behaviours, p.2, lines 110-127

Lines 123-141 have been updated to reduce redundancy. We hope that this clarifies and avoids repetition.

  • Table 1 is really results and should be placed there. Additional notation needed to state numbers, eg, sleeping status: n (%).

Per reviewer suggestion, we have moved Table 1 into the results section and added notation.

  • Tables 2 & 3 need some additional notation – assume they are SDs being reported? Also numbers and %?

Per reviewer suggestion, additional notation has been added in Table 2& 3.

  • Is fluid recorded in mls, although seems a lot? 32226 = 32,226, Please check all numbers in the table!

Thanks for your comment. The data on 24-hr total fluid consumed have been corrected in Table 3.

  • Looking at the numbers on p.6, lines 175-181, looks like the numbers in table 3 are missing some decimal points?

Per your comments, the numbers are reported in the test have revised to inconsistent with Table 3.

  • some of the findings were presented in methods and need moving to results.

Per reviewer suggestion, Table 1 and findings have been moved to the results section and in lines 158-159.

  • 7, lines 196-200: not quite sure what is being said here – please clarify.

We have revised lines 210-226. We hope this has clarified.

  • Section 4.1: this section seems a little muddled in places. Suggest rephrasing, and focusing on the title, which is fluid intake and physiol hydration status

Per reviewer suggestion, we have rephrased this section in lines 210-226.

  • Finer detail required to describe the studies being used in comparison in sections 4.1 and 4.2, to know whether like is being compared with like, and what would account for possible differences.

Per reviewer suggestion, more information has been added in lines 210-228.

  • I am not sure how the conclusion presented in section 4.2, line 229 links with the data, could the authors clarify?

This section has been updated and references made to our data that support our claims.

  • Further discussion of limitations needed, eg, why do they think their sample may have been well-hydrated?

From the data on 24-hour total fluid intake in Table 3, our average daily fluid intake for men was 3222ml and for women was 3303ml; the US National Academy of Medicine (NAM) recommends adequate total daily fluid intake for adult men and women as 3.7L/day and 2.7L/day, respectively. This means 61.5 % of our sample met, or exceeded, recommendations. Text has been altered in this section as well.  

  • How does this study inform our understandings and healthcare?

We appreciate the critical review and hope that our revision is seen as appropriate and improves the paper. We have added to our conclusion in hopes of providing clinical relevance of our study.

Reviewer 2 Report

Comments and Suggestions for Authors

Congratulations on a well-crafted article. The significance of the study is aptly introduced, and the content is both insightful and original. We have several suggestions aimed at enhancing the clarity and coherence of your article.

  1. Clarification of Terminology (Lines 27 to 39 and throughout the article): The distinction between "total fluid intake" and "fluid drank" or "liquids intake" or "drinks intake" needs elucidation to prevent confusion among readers. It is imperative to establish clear terminology for these concepts early on and consistently employ it throughout the article. This will ensure clarity for future readers, particularly when references are made to "fluid intake."
  2. Specificity in Methodology (Line 96): When referring to "Diet & Fluid Consumption Logs," please provide details regarding the tool used and include references for the procedural methodology.
  3. Clarity in Data Collection (Lines 101 to 108): Specify the timeframe for the collection of 24-hour urine specific gravity, and provide references for the analysis procedure of USG and urine osmolality. Additionally, offer insights into the tools utilized for measurement.
  4. Void Frequency Methodology (Section 2.2): Ensure clarity by explaining the methodology for void frequency assessment and include references for procedural details.
  5. Consolidation of Sections (Sections 2.3 and 2.4): Merge Section 2.3 "Health-related Behavior Recording and Assessment" with Section 2.4 to streamline the discussion. Additionally, provide insight into the rationale behind the development of this methodology, citing relevant literature to support its validity or acknowledging its development based on empirical evidence or common sense, with appropriate references. It should be considered as a limit of the study.
  6. Table Clarification (Table 3, 6, and 7): Table 3's figures for "24h total fluids consumed" may require verification, as they seem to yield quantities exceeding 30 liters per subject per day. Consider adding data on daily drinks consumed, as this information is pertinent, especially given the normohydrated population studied. Tables 6 and 7 should be accompanied by explanatory comments to aid future readers in interpreting the results, possibly necessitating additional elaboration in the statistical analysis section.
  7. Precision in Terminology (Lines 226, 227): Specify "total fluid intake" for clarity. Additionally, consider the potential impact of including data on the quantity of drinks consumed by subjects, as this may enhance the association with hydration knowledge questionnaire responses, particularly given the questionnaire's focus on drink quantity, and not on total fluids intakes. This suggestion is open for your consideration.

We believe implementing these suggestions will strengthen the coherence and comprehensibility of your article, facilitating its understanding and impact among readers.

Reviewer 3 Report

Comments and Suggestions for Authors

Authors investigated and found the association of self-reported fluid intake with fliud status but not with hydration knowledge nor with health habits.

The research itself was conducted appropriately, however, the objectives and the results of the study are of little clinical impact or importance.

* Fluid status and its variations assessed in this study were almost in the normal range. Thus, the high or low fluid status should not be related to clinical impact.

* I cannot understand why authors try to investigate healthy behaviors with fluid status. They should offer the reasons or previous reports why they should investigate the associations of the two variables.

* What is the clinical impact of the study? Even if the results are negative (except fluid intake with fluid status, which is totally natural and without questions), the non-associations of fluid status within normal limits with health knowledge or behaviors do not necessarily mean that it is the case when fluid status is abnormal.

* Previous report which authors inferred has shown the fluid status is not associated with hydration knowledge but with fluid intake attitude/behaviors. Why did authors adopt hydration knowledge but not fluid intake attitude/hehaviors?

Round 2

Reviewer 3 Report

Comments and Suggestions for Authors

No further comments